# Direct synthesis of a semiconductive double-helical phosphorus allotrope in a metal-organic framework

Sergei A. Sapchenko [1] ✉, Rodion V. Belosludov [2], Inigo J. Vitoria-Irezabal [1], Ivan da Silva [3], Xi Chen[1,4], George F. S. Whitehead [1], John Maddock[1], Louise S. Natrajan [1], Meredydd Kippax-Jones[1], Dukula De Alwis Jayasinghe[1], Carlo Bawn[1], Daniil M. Polyukhov [1], Yinlin Chen [1], Vladimir P. Fedin[5,6], Sihai Yang [1,7] ✉ & Martin Schröder [1] ✉

There remains much ambiguity regarding the structure of red phosphorus. We report the adsorption and photo-polymerisation of $P_4$ molecules encapsulated in an indium(III)-based metal-organic framework to afford a double-helical chain composite comprising of $[P_8]$ units. The similarity between the Raman spectrum of bulk red phosphorus and of the metal-organic framework – $(P_8)_n$ adduct suggests the presence of such helical chains in the structure of amorphous red phosphorus. This provides crystallographic evidence of the structural building blocks of the red phosphorus allotrope stabilized within the pores of a metal-organic host. The $(P_8)_n$ inclusion compound is an air-stable semiconductor with a band gap of 2.3 eV, which is relevant for gas detection and photo-catalysis. We demonstrate that this phosphorus adduct demonstrates a 10-fold increase in conversion in the oxidation of methyl orange dye compared with the parent metal-organic framework material.

Phosphorus is remarkable for the structural diversity of its allotropes, but only a few of them are chemically stable and possess desired properties. The molecular forms of low nuclearity species are unstable, with white phosphorus $P_4$ self-igniting in air, and $P_2$ and $P_6$ species existing only in the gas phase at high temperature[1,2]. In contrast, polymeric red, violet and black phosphorus can be handled readily under ambient conditions. Phosphorene, a monolayer of black phosphorus, was recently demonstrated to be a semiconductor with a tuneable bandgap[3,4]. However, these sheets of black phosphorus are not stable towards a combination of moisture, oxygen and light. Another drawback is the high energy consumption during the synthesis of this allotrope from $P_4$ at 1.3 GPa and 200 °C[5]. Another important

and stable allotrope is red phosphorus, which is more stable than black phosphorus with a wide bandgap of up to 2.4 eV[6]. Decreasing the bandgap of red phosphorus-based materials is a challenging task, and extensive theoretical and synthetic investigations have been conducted to develop new polymeric forms of phosphorus with improved stability and physiochemical properties.

Von Schnering and Baudler have demonstrated[7,8] various types of clusters of phosphorus, including the structural elements of black phosphorus. *Cis*- and *trans*-linear chains, helical and double-helical chains containing $[P_8]$ cages directly bound to each other have been predicted recently to be more stable than $P_4$[9]. A phase-pure sample of fibrous phosphorus featuring parallel chains of $P_8$ building blocks[10] has

[1]Department of Chemistry, University of Manchester, Manchester M13 9PL, UK. [2]Institute for Materials Research, Tohoku University, Sendai 980-8577, Japan. [3]ISIS Facility, STFC Rutherford Appleton Laboratory, Oxfordshire, Oxfordshire OX11 0QX, UK. [4]College of Chemistry and Chemical Engineering, China University of Petroleum (East China), Qingdao 266580, PR China. [5]Nikolaev Institute of Inorganic Chemistry SB RAS, 3 Lavrentiev Ave, Novosibirsk 630090, Russian Federation. [6]Faculty of Natural Sciences, Novosibirsk State University, 1 Pirogov Str., Novosibirsk 630090, Russian Federation. [7]College of Chemistry and Molecular Engineering, Beijing National Laboratory for Molecular Sciences, Peking University, Beijing 100871, PR China.
✉e-mail: ssapchenko@gmail.com; Sihai.Yang@manchester.ac.uk; M.Schroder@manchester.ac.uk

been isolated by heating ultra-pure red phosphorus in the presence of $CuCl_2$ in vacuo[11]. Supramolecular chemistry can promote controllable polymerisation and stabilisation of phosphorus chains within the nanochannels of a porous host. Thus, CuI has been used to stabilise phosphorus nanorods in the form of $(CuI)_8P_{12}$ and $(CuI)_3P_{12}$ adducts[12], as well as confining and stabilising anionic $P_{12}{}^{2-}$ and $P_{14}{}^{2-}$ species[13]. Activated carbons and carbon nanotubes can adsorb tetrahedral $P_4$ molecules[14], with the latter also able to act as a host for the poly-merisation of encapsulated white phosphorus molecules into single-stranded chains consisting of butterfly $P_4$ fragments[15–17]. Due to the amorphous nature of these materials, their structure determination remains elusive.

A rich host–guest chemistry[18,19], tuneable porosity, high crystal-linity and potential stability of metal-organic frameworks (MOFs) make them ideal hosts for the inclusion of white phosphorus molecules, which might be further polymerised using visible light and analysed structurally by diffraction[20,21]. Moreover, the doping of MOFs with phosphorus chains of atomic width may improve the semiconducting properties of the resulting P–MOF composites. Here, we report the synthesis of a double-helical phosphorus chain comprising butterfly $P_8$ monomers stabilised in the helical channels of an indium-based MOF, MFM-300(In). The similarity between Raman spectra of bulk red phosphorus and the obtained adduct suggests the presence of such helical chains within the structure of amorphous red phosphorus, thus representing a comprehensive crystallographic study on the structural building units of the red phosphorus allotrope.

## Results

### Synthesis of $P_4$@MFM-300(In)

MFM-300(In), $[In_2(OH)_2(bptc)]$ (bptc$^{4-}$ = 3,3′,5,5′-biphenyltetracarbo xylate)[22], was chosen for the polymerisation of white phosphorus molecules as it features non-intersecting cylindrical channels of ca. 8 Å in diameter. These are ideal to accommodate the white phosphorus $P_4$ precursor and resulting polymer. Additionally, MFM-300(In) crystals are chemically stable and colourless to enable visible light-induced polymerisation. The adsorption of white phosphorus into activated MFM-300(In) was performed in the dark. Single crystal X-ray diffrac-tion (SCXRD) analysis confirms that $P_4$@MFM-300(In) crystallises in the tetragonal space group $I4_122$ with similar unit cell parameters [$a$ = 15.5333(1) Å, $c$ = 12.3260(1) Å, $V$ = 2974.06(4) Å$^3$] to the pristine framework. The channels are filled with tetrahedral $P_4$ molecules (Fig. 1), which occupy one of three crystallographically independent sites, denoted $P_4$(I), $P_4$(II) and $P_4$(III). The P–P distances lie in the range 2.1959(1) – 2.2277(1) Å for $P_4$(I), 2.1956(1) – 2.2276(1) Å for $P_4$(II), and 2.1954(1) – 2.2274(1) Å for $P_4$(III), consistent with those reported for single P–P bonds[18]. Each site has a distinct binding mode. $P_4$(I) is aligned towards the bridging OH-group within the channel with a O−H...P hydrogen bonding distance of 2.794(5) Å. The $P_4$ molecule is favourably positioned between two phenyl rings of the ligands with close P...π-$C_6H_6$ contacts of 3.189(6) and 3.462(1) Å. There is also an interaction between P and the carboxylate oxygen atoms $O_{COO}$ [P...$O_{COO}$ = 3.718(3) Å] of the host. $P_4$(II) molecules and their disordered counterparts $P_4$(III) reside towards the centre of the pore. The shortest contacts between $P_4$(II) and the framework are from the interaction of P with two carboxylic oxygen atoms $O_{COO}$ [P...$O_{COO}$ = 3.565(7), 3.550(1) Å]. The two phenyl rings are located further from the P atoms with P...π-$C_6H_6$ distances of 4.118(1) and 4.33(2) Å. $P_4$(III) molecules show multiple interactions with the carboxylate groups [P...$O_{COO}$ = 3.48(1), 3.55(2), 3.58(2), 3.84(3), 3.98(1) Å]. Due to translational symmetry, all of the $P_4$ sites (I–III) result in a variety of possible posi-tions of $P_4$ within the channels of MFM-300(In) (Fig. 1b and Supple-mentary Fig. 3). Elemental analysis and TGA data (Supplementary Fig. 2) confirm the total concentration of $P_4$ molecules corresponds to the formula $[In_2(OH)_2(bptc)]\cdot2.5P_4\cdot H_2O$, which also matches the structural data. The phase purity of the bulk sample was confirmed by

powder X-ray diffraction (PXRD, Supplementary Fig. 5). Scanning electron microscopy (SEM) coupled with energy dispersive X-ray (EDX) mapping confirmed a homogenous distribution of phosphorus within the crystals and an absence of aggregation of phosphorus particles on the surface of the crystals (Fig. 2c and Supplementary Fig. 13). Interestingly, in contrast to white phosphorus, $P_4$@MFM-300(In) is comparatively easy to handle as it does not self-ignite in air.

Periodic DFT calculations using the Vienna Ab Initio Simulation Package (VASP), with correction for Van der Waals exchange-correlation error, were performed to gain further insights into the structure of these materials[23,24]. The optimised positions of the $P_4$ molecules were found to be in agreement with the crystal structure, and all the resulting adducts were found to be energetically favourable. The calculations confirm that at low concentrations of $P_4$ (up to 1 $P_4$ molecule per In centre), $P_4$(I) is the preferred adsorption site (Sup-plementary Fig. 8), and at high loadings, adsorption at sites $P_4$(II) and $P_4$(III) occurs. The saturated material with a loading of 2 $P_4$ molecules per formula unit is energetically favourable (total energy $E$ = −23.6836 eV) (Supplementary Fig. 8a) and corresponds to the experimental uptake. In addition, charge distribution calculations indicate a depletion of electron density on the guest phosphorus molecules. The partial charge of the phosphorus atoms, $P^{\delta+}$, (see SI Section 5.3 and Source data file), rises as high as +0.11$e$, indicating a charge transfer between $P_4$ and the framework via strong host–guest interactions (Fig. 2a).

### Conversion of $P_4$@MFM-300(In) to $(P_8)_n$@MFM-300(In)

$P_4$@MFM-300(In) was irradiated with a 300 W Xe light source ($\lambda$ = 350−760 nm) to induce the polymerisation of the $P_4$ species within the channels. Within 6−8 h (depending on the amount of starting material), the pale yellow crystals turned deep orange with full reten-tion of crystallinity (Supplementary Figs. 6, 7). SEM and EDX show a homogenous distribution of phosphorus within the samples, with full retention of morphology and the absence of phosphorus particles on the external surfaces after the polymerisation (Fig. 2c). Elemental analysis confirms a concentration of phosphorus of 30 wt% in the polymerised adduct, comparable with the initial concentration of phosphorus in $P_4$@MFM-300(In) (34 wt%). $N_2$ adsorption measure-ments at 77 K confirm full occupation of the pore space in $(P_8)_n$@MFM-300(In) since it is not porous to $N_2$. This is in contrast to bare MFM-300(In), which shows a BET surface area of 1000 m$^2$ g$^{-1}$ (see SI Section 10). SCXRD analysis confirms the integrity of the host framework with no apparent structural change in the MOF platform. In contrast, the structure of guest phosphorus changes significantly on irradiation, with four crystallographically independent phosphorus atoms forming connected butterfly $P_4$ fragments of $C_{2v}$ symmetry. These are similar to the $P_4$ fragments that have been observed in coordination and orga-nophosphorus compounds[25,26]. The distribution of P–P bond distances is slightly distorted in the fragment, and ranges from 1.934(5) to 2.243(5) Å. The fragments are aggregated into a chain of elemental phosphorus running along the $c$-axis and can be viewed as connected $P_8$ clusters (Fig. 3a and Supplementary Fig. 5). The same tubular structure was observed independently by refining high-resolution PXRD data from the powder sample of $(P_8)_n$@MFM-300(In) (Supple-mentary Fig. 6 and Supplementary Tables 4, 5) in which the refined $P_n$ moieties show a similar variation of P–P bond length [2.0(2) - 2.5(2) Å]. The P(3)P(2)P(4), P(2)P(4)P(3′) and P(3′)P(3)P2 angles are 74(2)°, 96(2)°, and 101(2)°, respectively. These $P_4$ moieties are chemically bound to each other to form a unique $P_8$ building block which forms a double helix chain running along the $c$ axis. These cluster and double-helical chains are unusual in the structural chemistry of elemental phosphorus (Fig. 3d and Supplementary Fig. 4). Interestingly, the P–P bond lengths within helices of butterfly $P_4$ subunits are similar [1.99(5) – 2.01(5) Å], while the P–P distance between two $P_4$ subunits in the $P_8$ building block is substantially longer [2.24(5) Å, See Supplementary

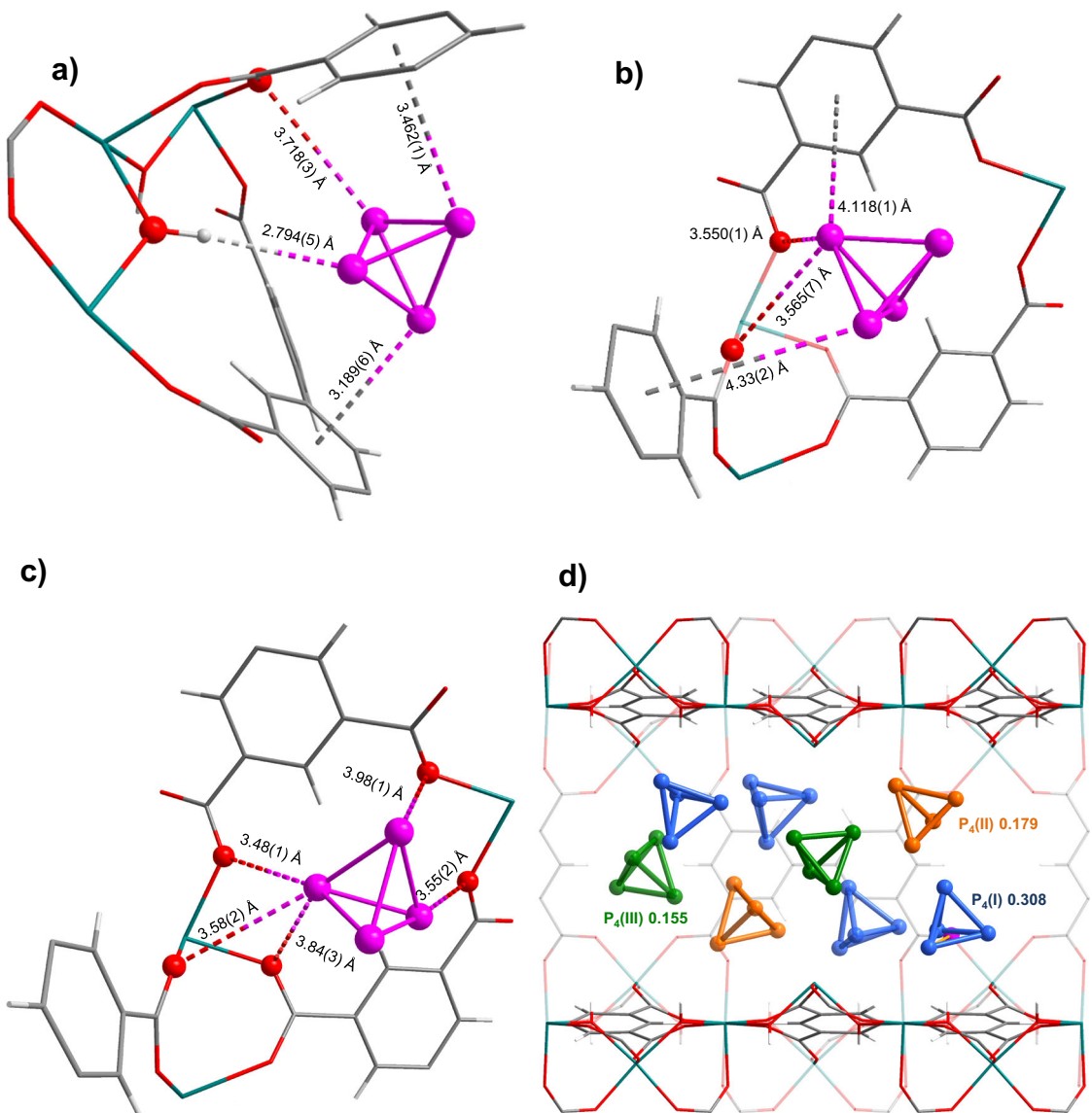

**Fig. 1 | Views of the structure of P₄@MFM-300(In) and binding modes of P₄ in P₄@MFM-300(In). a** Site $P_4(I)$; **b** Site $P_4(II)$; **c** Site $P_4(III)$. The shortest contacts are marked with dashed lines. Indium atoms are shown green, C – grey, O – red, P – pink, H – pale grey; **d** Packing of $P_4$ molecules within the channels of MFM-300(In): view in ($\bar{1}10$) plane. $P_4(I)$ molecules are shown blue, $P_4(II)$ – orange, $P_4(III)$ – green. The numbers indicate crystallographic occupancy.

Fig. 5]. This latter value is comparable with P–P bond length reported for sterically hindered diphosphanes [2.291(4)–2.357(2) Å][27] and in hypothetical small phosphorus cluster molecules [2.24 Å for one of possible $P_6$ clusters][28], indicating weaker bonding. To confirm that the observed double helix is not a mere result of the structural disorder of $P_4$ subunits and to explain the deviation in P–P–P angles, we used ab initio DFT calculations to analyse the system (See Methods and SI Section 5 for details). The analysis confirms that single chains of butterfly $P_4$ fragments, which have been thought to be the main components of red phosphorus since the 1950s[29], collapse within the channels of the host MOF framework. The resultant double helix $[P_8]_n$ polymer (Fig. 3b and Supplementary Figs. 10, 11) is thermodynamically stable with a total energy for its optimised form of −16.62718 eV (−0.51960 eV per phosphorus atom).

A distortion in each of the $P_4$ subunits within the cluster can be rationalised by considering the redistribution of electron density and charge transfer between the double helix and the framework. DFT calculations performed on the optimised $(P_8)_n$@MFM-300(In) fragment reveal a depletion of electron density on the phosphorus atoms and charge transfer between the phosphorus chain and the functional groups of the host framework (Fig. 2a). The partial charges can reach +0.092e (see SI Section 5.3 and Source data file). The depletion of the electronic density in the $P_4$ subunit, as well as steric hindrance, explains the flattening of the fragment in comparison to known neutral organophosphorus compounds featuring the same fragment. Indeed, it has been shown previously that electron depletion can cause a flattening of the $P_4$ fragments, for example, in the cationic form $\{(Me_2P^+)_2(t\text{-}Bu_2P)_2\}$ where the $P_4$ cluster is significantly flattened with a fold angle for the $P_4$ scaffold of 139.4°[30]. Binding to metal centres and emergence of ligand to metal charge transfer, especially within a sterically hindered geometry, may lead not only to flattening, but also to aromatisation within a cyclic $P_4$ moiety[31–34].

**Spectroscopy**

The formation of the double helix chain has also been tracked by Raman spectroscopy. The Raman spectrum of P₄@MFM-300(In) features three intense peaks at 361, 467, and 601 cm⁻¹ corresponding to Raman-active $F_2$, $E$, $A_1$ vibrations of the $P_4$ tetrahedron, respectively (Supplementary Fig. 14a)[35]. On polymerisation, new peaks in the Raman spectrum appear at 357(m), 381(m), 409(w), 456(s),

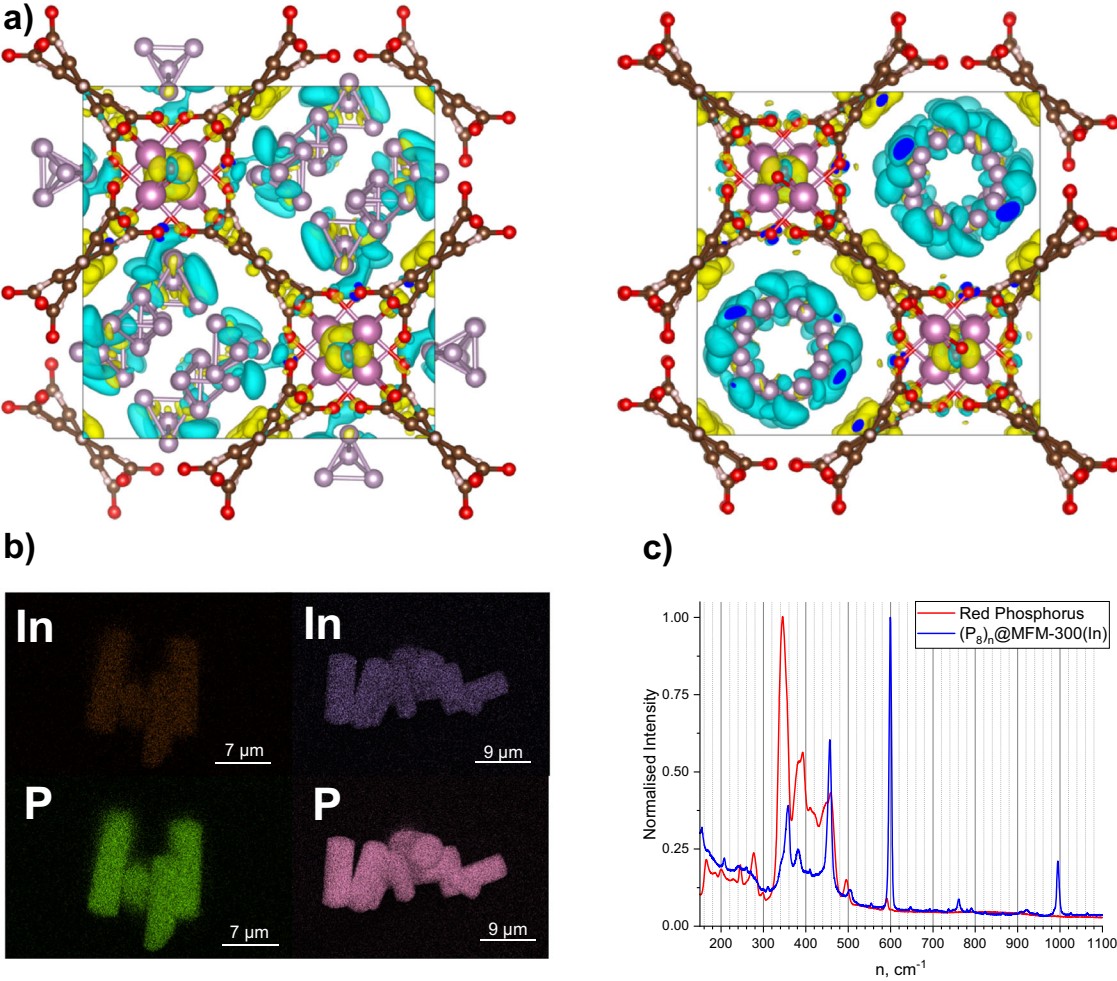

**Fig. 2 | Comparative physicochemical characterisation of P₄- and photopolymerised (P₈)ₙ@MFM-300(In). a** Charge-density isosurfaces for the interaction of MFM-300(In) framework with (left) $P_4$ molecules; (right) double helices of $(P_8)_n$. Yellow represents the accumulation and blue the depletion of the electron density. Indium atoms are shown pink, C – brown, O – red, P – pale violet, H – pale brown. The DFT results are visualised using the VESTA code; **b** EDX mapping images of In and P in $P_4$@MFM-300(In) (left) and $(P_8)_n$@MFM-300(In) (right); **c** comparison between Raman spectra of commercial red phosphorus (red), and $(P_8)_n$@MFM-300(In) (blue).

501(w), and 599(s) cm⁻¹ (Fig. 2c). These match the corresponding peaks of amorphous red phosphorus at 345(s), 382(m), 410(w), 458(m), 496(w), and 591(w) cm⁻¹ (Fig. 2c), and suggest the presence of such helical chains in red phosphorus. It is worth noting that both spectra are very different from the predicted Raman spectra of any single chain built up from $P_4$ fragments[36]. Thus, the obtained double helix structure observed here is consistent with one of the important components of amorphous red phosphorus. This is distinct from a predicted molecular structure of amorphous red phosphorus[37] and experimentally observed double helices within carbon nanotubes[38]. The similarity between red phosphorus and photopolymerised species in $(P_8)_n$@MFM-300(In) can be observed by NMR spectroscopy as well (See SI Section 8). The solid-state ³¹P NMR spectrum of $(P_8)_n$@MFM-300(In) shows one broad signal at –120 ppm, which is close to one of the peaks observed in the NMR spectrum of pure red phosphorus (Supplementary Fig. 15). The Raman spectra of freshly prepared and 3-month-old $(P_8)_n$@MFM-300(In) demonstrate no significant differences (Supplementary Fig. 14b), confirming good general stability of the material toward prolonged exposure to air. A weak signal at 924 cm⁻¹ is present in the spectra of fresh and air-exposed samples and is characteristic of the presence of P−O bonds[39]. However, the consistently low intensity of this peak in both spectra suggests that this represents only surface oxidation.

To gain further understanding of the electronic structure of these compounds, we determined the bandgap of MFM-300(In), $P_4$@MFM-300(In) and $(P_8)_n$@MFM-300(In) by UV-Vis spectroscopic experiments (see SI Section 9). Upon adsorption of $P_4$ and its polymerisation, the bandgap decreases from 3.9 in MFM-300(In) to 3.2 in $P_4$@MFM-300(In) to 2.3 eV in $(P_8)_n$@MFM-300(In) (Table S6). This is a narrower bandgap than in most guest-free MOF complexes such as MIL-101, ZIF-8, HKUST-1 and MOF-74, and is comparable with other well-performing semiconducting MOF-based host–guest systems (Table S7). The decrease in bandgap in going from MFM-300(In) to $P_4$@MFM-300(In) to $(P_8)_n$@MFM-300(In) is explained by partial density of states (PDOS) calculations (Supplementary Fig. 12). In short, the narrowing of the bandgap in this system is caused by the contribution of phosphorus orbitals to the valence band maximum (VBM) and the conduction band minimum (CBM), which is the largest in the case of $(P_8)_n$@MFM-300(In). In guest-free MFM-300(In), the VBM and CBM are affected primarily by the orbitals on the bridging ligand.

The decrease in bandgap in the composite $(P_8)_n$@MFM-300(In) affects a number of physical properties compared to the bare MOF, allowing $SO_2$ detection by monitoring changes in the dielectric constant of the material (see SI Section 11.1). Importantly, the phosphorus adduct has a much higher ability to generate a photo-current (SI Section 11.3, Supplementary Fig. 22), and is luminescent (SI Section 12,

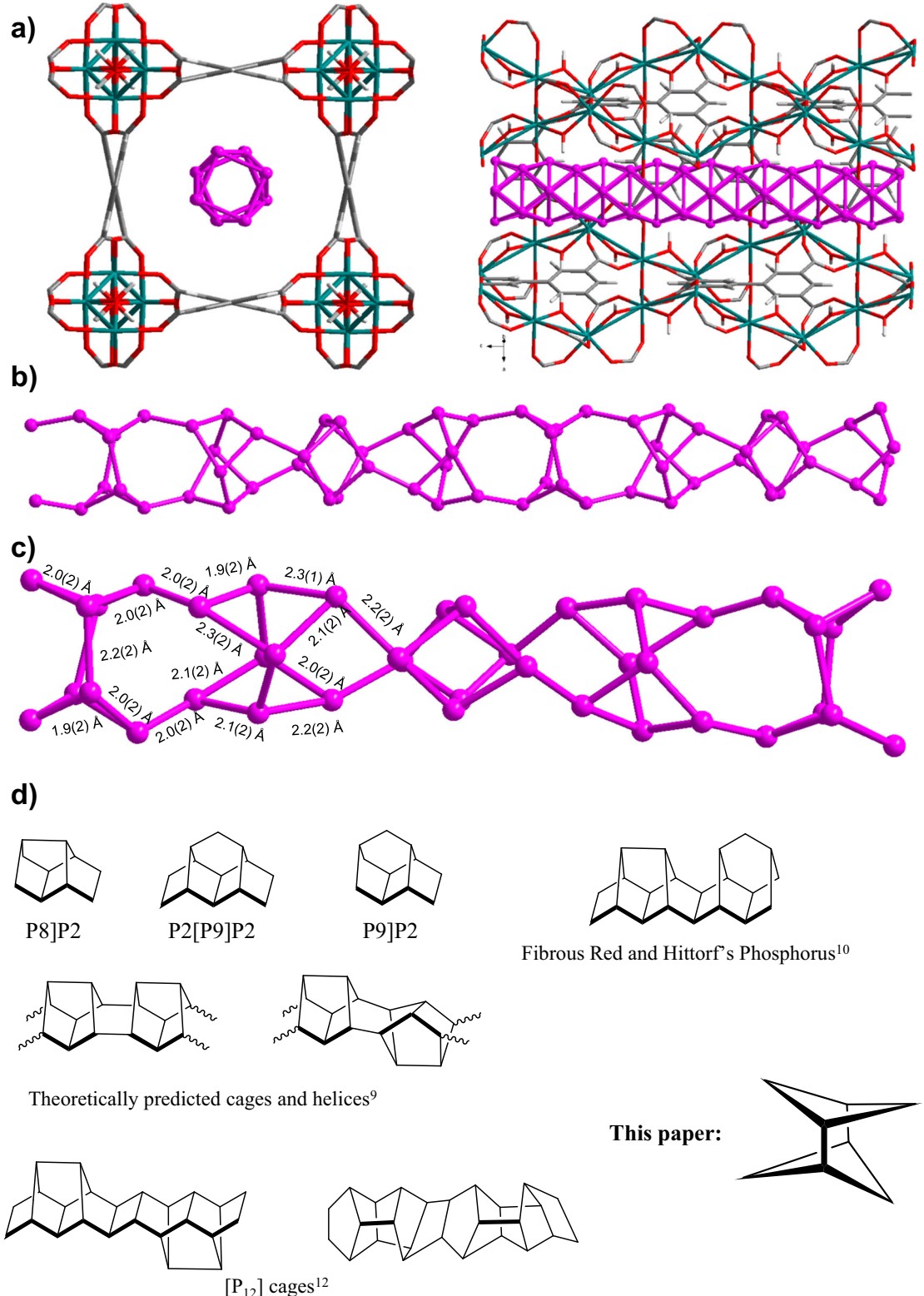

**Fig. 3 | Structural studies of (P₈)ₙ@MFM-300(In) adducts. a** The packing of the disordered nanotube of phosphorus atoms within the channels of MFM-300(In): view along *c* axis (left*)* and in (1̄10) plane (right). Indium atoms are shown green, C – grey, O – red, P – pink, H – pale grey; **b** View of the DFT-optimised double helix; **c)** view of the fragment of the double helix refined from powder diffraction data; **d** Comparison of P₈ unit with other Pₙ cluster fragments predicted or experimentally observed in polymeric phosphorus allotropes.

Supplementary Fig. 23), key features for photo-catalysis and for water purification[40]. To examine this effect, guest-free MFM-300(In) and (P₈)ₙ@MFM-300(In) were tested for the photocatalytic degradation of rhodamine B and methyl orange as model compounds of water

pollutants. In each test, the concentration of the catalyst and the pollutant was 0.5 and 0.01 g L⁻¹, respectively. No additional oxidant other than air was used. Prior to irradiation with an incandescent halogen lamp (250 W), the catalyst was dispersed in the solution for 30 min to reach

**Table 1 | Selected examples of photocatalytic degradation of dyes by MOFs and P-containing materials**

| Catalyst | Catalyst concentration g/L | Dye* | Oxidant | Time | Conversion % | |
|---|---|---|---|---|---|---|
| P$_8$-MFM-300 | 0.5 | 10 mg/L RhB | Air | 150 min | 90 | This work |
| P$_8$-MFM-300 | 0.5 | 10 mg/L MO | Air | 300 min | 90 | This work |
| Milled red phosphorus (RP-36) | 0.4 | 10 mg/L RhB | Air | 6 hours | 88 | 46 |
| Bulk black phosphorus | 1.0 | 2·10$^{-4}$ mg/L MO | Air | 120 min | 25 | 62 |
| Black phosphorus nanosheets @ Graphene Oxide | 1.0 | 2·10$^{-4}$ mg/L MO | Air | 120 min | 100 | 62 |
| nMLM | 0.1 | 50 mg/L RhB | 1 ml H$_2$O$_2$ | 240 min | 92 | 63 |
| MIL-53-Fe | 0.01 | 4·10$^{-4}$ M MB | Air | 1 h | 11 | 42 |
| UiO66 | 0.1 | 20 mg/L MO | Air | | 0 | 41 |
| UiO66-1.25Ti | 1.0 | 10 mg/L MB | Air | 80 min | 82 | 64 |
| HKUST-1 | 0.125 | 25.28 mg/L RB13 | Air | 14 h | 40 | 65 |
| MIL-100 (Fe) | 0.3 | 5 ppm MO | Air | 7 h | 64 | 43 |
| Au@MIL-100(Fe) | 0.125 | 0.02 g/L MO | 10 mM H$_2$O$_2$ | 1.57 h | 100 | 66 |
| (Fe-Ti)-(MOF-NH2) (3:1) | 0.1 | 0.05 g/L OrangeII | 14 mM H$_2$O$_2$ | 10 min | 100 | 66 |
| (Fe$_3$O$_4$)-(MIL-88B (Fe)) | 0.2 | MB/ RhB 0.1:0.01 mM | 20 mM H$_2$O$_2$ | 1.33 h | 100 | 66 |
| MIL88(Fe) | 0.4 | 10 mg/L RhB | 20 mM H$_2$O$_2$ | 80 min | 50 | 67 |
| Bi$_2$O$_3$ – Cu-MOF | 1.0 | 10 mg/L RhB | Air | 60 min | 44 | 44 |
| Bi$_2$O$_3$ – Cu-MOF– Graphene Oxide (BCG-2) | 1.0 | 10 mg/L RhB | Air | 60 min | 80 | 44 |

*RhB rhodamine B, MO methyl orange, MB methylene blue

adsorption equilibrium. The reaction rate was monitored by the UV-Vis spectroscopy (Supplementary Figs. 24, 25). As expected, the inclusion of the phosphorus chain within MFM-300(In) boosts the catalytic performance. Thus, (P$_8$)$_n$@MFM-300(In) promotes the oxidation of Rhodamine B with a 90% conversion of a substrate after 150 min of irradiation, while the reaction with MFM-300(In) shows a conversion of just 56% after 270 min. The use of red phosphorus resulted in only 10.6% conversion after 150 min. Moreover, bare MFM-300(In) catalyses the oxidation of methyl orange poorly, with a conversion of only 9.6% after 300 min, whereas (P$_8$)$_n$@MFM-300(In) demonstrates a tenfold increase in conversion to 90.3% in 300 min. Bulk red phosphorus is also very efficient in this reaction, demonstrating 99% conversion after 300 min. The heterogenous nature of catalysis was confirmed by filtration tests (Supplementary Fig. 26), in which the solid catalyst is removed and subsequent catalysis monitored. The stability of the catalyst was monitored and confirmed by PXRD measurements (Supplementary Fig. 27). Overall, the introduction of polymeric phosphorus significantly improves the photocatalytic performance of the resulting material, showing superior catalytic performance to MOF materials such as UiO66[41], MIL-53[42], MIL-100[43], and bismuth oxide-MOF adducts[44]. High conversion rates are often reported when hydrogen peroxide is used as an oxidant (Table 1). However, the use of hydrogen peroxide has several disadvantages compared to air, as H$_2$O$_2$ is an aggressive oxidant and may cause explosions and equipment failures when used in industrial scale[45]. The conversion of Rhodamine B using (P$_8$)$_n$@MFM-300(In) is especially effective, and a prepared sample of milled red phosphorus[46] was able to reach 88% conversion of Rhodamine B, but in double the length of time compared with (P$_8$)$_n$@MFM-300(In). Table 1 summarises data on the photo-degradation of dyes using for (P$_8$)$_n$@MFM-300(In) and other MOFs and P-containing materials.

## Discussion
In summary, the encapsulation and photo-polymerisation of guest white phosphorus molecules in P$_4$@MFM-300(In) have been investigated. The high crystallinity of the adduct allowed the determination of the structure of P$_4$@MFM-300(In) and of the chain allotrope (P$_8$)$_n$@MFM-300(In), the latter demonstrating an unusual helical shape stabilised within the pores of the MOF. Raman spectroscopy confirms that the helical phosphorus chain observed in (P$_8$)$_n$@MFM-300(In)

shows similar characteristics to that of amorphous red phosphorus. The confinement of phosphorus chains in MFM-300(In) significantly changes its functional properties, and the resultant composite is an air-stable semiconductor with a bandgap of 2.3 eV, with potential for gas detection and photocatalytic applications. This methodology to stabilise otherwise unstable species within a porous host can potentially be extended to other porous OH-decorated MOFs, since P$_4$ molecules strongly interact with hydroxyl groups, as demonstrated here. Other small molecules based upon S, Se, As and Sb might also be used as substrates for MOF-assisted photo-polymerisation, which will be a powerful way to modify and control the electronic structure and photocatalytic performance of the resulting encapsulated and protected adducts, aggregates and polymers.

## Methods
### Materials and characterisation techniques
Indium(III) nitrate hydrate (Aldrich, 99.9%), biphenyl-3,3′,5,5′-tetra-carboxylic acid (TCI, 98%), DMF (Fisher Chemical, 99.8%), acetonitrile (Fisher Chemical, 99.5%), concentrated nitric acid (Fisher Chemical, 67–70 wt%), red phosphorus (Aldrich, 97%), Rhodamine B (Fisher Chemical, 95%), methyl orange (Aldrich, 95%), sodium sulphate (Aldrich, 99.0%), Nafion D-521 (Fisher Chemical, 5 wt% dispersion in water and 1-propanol, 0.92 meq g$^{-1}$ exchange capacity) were used as purchased. White phosphorus was obtained from red phosphorus[47]. Elemental analysis was performed on a Flash 2000 elemental analyser, and the morphology of the crystallites was determined on a SEM on a Quanta 650 microscope. A Shimadzu UV-2600 spectrometer was used to collect solid-state UV-Vis spectra in the range from 800 to 200 nm, and solid-state luminescence excitation and emission spectra recorded on an Edinburgh Instrument FP920. Solid-state $^{31}$P NMR spectra were collected on a Bruker AVIII 400 instrument, and TGA plots (Supplementary Fig. 2) were collected on a Perkin Elmer Pyris1 thermogravimetric analyser under N$_2$ at a flow rate of 100 mL/min and heating rate of 5 °C /min. Raman spectra were collected on a Horiba Scientific Xplora plus Raman Microscope using a 785 nm laser with a grating of 1200 gr mm$^{-1}$. A total of 10 acquisitions with an exposure time of 10 s were used to collect spectra. A 300 W Xe lamp was used at a wavelength of 350–760 nm (visible output 5000 lm) for irradiation of P$_4$@MFM-300.

## Synthesis of MFM-300(In)

MFM-300(In) was obtained according to a modification of the published method[22] as follows: a mixture of In(NO$_3$)$_3$·5H$_2$O (0.390 g, 1.0 mmol), biphenyl-3,3′,5,5′-tetracarboxylic acid H$_4$L (0.033 g, 0.1 mmol), DMF (10 ml) and acetonitrile (10 ml) was acidified with 60 drops of concentrated HNO$_3$. The resulting slurry was sealed in a pressure flask, sonicated in an ultrasonic bath and heated at 80 °C for 3 days. This yielded a highly crystalline material, which was filtered-off, washed five times with DMF and dried on air. The sample was activated by soaking in acetone for a week with subsequent heating in a dynamic vacuum at 120 °C overnight. Yield 0.041 g (70%, based on H$_4$L).

## Synthesis of P$_4$@MFM-300(In)

CAUTION: White phosphorus is highly toxic and self-ignites in air. All the operations with white phosphorus must be performed under an inert atmosphere using the Schlenk technique. Due to the light-sensitivity of the target compound and of P$_4$, the reaction should be carried in darkness.

Activated guest-free MFM-300(In) powder was placed in a Schlenk vessel, which was connected to another Schlenk flask filled with white phosphorus (Supplementary Fig. 1). The setup was evacuated, the valves closed (without refilling the setup with inert gas), and the flask with the white phosphorus was heated on a water bath for 1 h. The setup was cooled and left at room temperature for 3 days in darkness to ensure maximum adsorption of phosphorus vapour by MFM-300(In). To remove excess condensed white phosphorus, the flask with MFM-300(In) was disconnected and connected to a clean receiving Schlenk flask. The setup was degassed again, and the flask with MFM-300(In) was heated in the water bath at 50 °C for 1–2 h, then cooled down and left overnight. The setup was refilled with N$_2$, and the receiving flask containing excess white phosphorus was disconnected. The obtained pale-yellow powder of P$_4$@MFM-300(In) was stored in the Schlenk vessel under an inert atmosphere in the dark (in a box or fully wrapped with tin foil) as the sample is light-sensitive even to normal daylight. Analysis (calcd., found for In$_2$C$_{16}$H$_{10}$O$_{11}$P$_{10}$): C (20.54, 20.94), H (1.29, 1.10), N (0, 0), P (33.10, 33.75). TGA ($\Delta m$, %): −2% (H$_2$O), −34% (2.5P$_4$).

## Synthesis of (P$_8$)$_n$@MFM-300(In)

A sample of P$_4$@MFM-300(In) (0.100 g, 0.109 mmol) was placed in a Petry dish and irradiated with a 300 W Xe lamp until all the samples turned deep orange. The powder was stirred every 2 h to ensure that all the sample was irradiated fully. The irradiation time depends on the amount of starting material but usually takes 6–8 h. The obtained orange material was transferred into a Schlenk vessel and heated in a dynamic vacuum overnight at 100 °C. Yield 0.098 g (90%). Analysis (calcd., found for In$_2$C$_{16}$H$_8$O$_{10}$P$_{9.7}$(H$_2$O)$_5$): C (19.60, 19.75), H (1.85, 1.40), N (0, 0), P (30.65, 30.38). TGA ($\Delta m$, %): −9% (5H$_2$O).

## Details of DFT calculations

First-principles calculations were performed within the framework of density functional theory (DFT), as implemented in the Vienna Ab initio Simulation Package (VASP version 5.4.4)[23,24]. Electron exchange-correlation was treated by the generalised gradient approximation (GGA) with Perdew, Burke, and Ernzerhof (PBE) parameterisation[48], and interactions between the ion cores and valence electrons were modelled by the all-electron projector augmented wave (PAW) method[49,50]. The plain-wave cutoff energy was 400 eV, and convergence in energy ($10^{-4}$ eV) and force ($3 \times 10^{-3}$ eV/Å) were used during the optimisation procedure. Brillouin zone integrations were performed using the Monkhorst-Pack k-point mesh[51] with the 4 × 4 × 5 and 2 × 2 × 1 grids for calculation of single cell and 1 × 1 × 3 supercell, respectively. In order to properly estimate the weak non-covalent interactions, the Grimme parameterisation was applied[52].

The adsorption energies ($E_{ads}$) were calculated as the difference between the sum of the binding energies of the empty MOF ($E_{host}$) and the number of non-coordinated guest P$_4$ clusters ($n \cdot E_{guest}$), where $n$ is the number of adsorbed non-coordinated P$_4$ clusters, and that of the adsorbed system ($E_{host+guest}$) and the adsorption state, with a negative value of $E_{ads}$ being thermodynamically favourable [Eq. (1)].

$$E_{ads} = E_{host+guest} - (E_{host} + n \cdot E_{guest}) \tag{1}$$

In the case of chain calculations, $E_{guest}$ was considered for a single chain configuration. The interaction between the porous host and the substrate has been demonstrated by a charge-density isosurface, and the difference in charge density (excess and depletion electrons) was estimated as:

$$\Delta\rho = \rho(host+guest) - \rho(host) - \sum_{k=1}^{n} \rho_k(guest) \tag{2}$$

and the obtained charge density isosurfaces for the guest−host interactions were visualised using the VESTA code[53].

The theoretical analysis described above has been verified in previous studies[54–57]. The effective charge of atoms was evaluated by using the Bader analysis algorithm[58–61].

## Mott-Schottky plot and photo-current measurements

Mott-Schottky plots were recorded on a CHI660E workstation (CH Instruments, USA) with a conventional three-electrode system using a 0.5 M Na$_2$SO$_4$ aqueous solution. Preparation of the working electrode: 4 mg of bulk red phosphorus, MFM-300(In) or (P$_8$)$_n$@MFM-300(In) were dispersed in a solution of 4 mL ethanol and 10 μL Nafion D-521 to generate a homogeneous slurry. About 10 μL of the slurry was transferred and coated onto glassy carbon (diameter of 3 mm) and then dried. An Ag/AgCl electrode was employed as the reference electrode, and a platinum plate was used as the counter electrode.

## Photocatalytic experiments

About 5.0 mg of the catalyst [(P$_8$)$_n$@MFM-300(In), guest-free MFM-300(In) or red phosphorus] was dispersed in 10 mL of the 0.1 g L$^{-1}$ aqueous solution of rhodamine B or methyl orange. After vigorous stirring in the dark for 30 min, the reaction mixture was irradiated with an incandescent 250 W halogen lamp for 300 min. The rate of the reaction was monitored by recording the UV-Vis spectra of the aliquots taken 30, 90, 150, 210, and 300 min after the beginning of the reaction. After completion of the reaction, the solid-state catalyst was separated by centrifugation at 11872 x $g$ for 15 min, washed multiple times with acetone and dried on air. For the filtration test, the catalyst was removed by filtration after 30 mins of irradiation and by centrifugation at 11,872×$g$ for 15 min. The remaining liquid phase was further irradiated for 270 min and analysed by UV-Vis spectroscopy. For blank tests, the catalyst and the dye solution taken in the same ratio as described above were stirred in complete darkness for 24 h. The concentration of dye in these solutions was monitored by UV-Vis spectroscopy.

## Data availability

TGA, single-crystal, powder X-ray and synchrotron diffraction analyses, details of DFT calculations, SEM data, Raman, SNMR, UV-Vis spectroscopy data, adsorption isotherms, electrochemical, photoluminescence, and photocatalytic experimental data generated in this study in the Supplementary Information. The crystallographic data of the materials reported in this work have been deposited in the Cambridge Crystallographic Data Centre (CCDC) under accession numbers CCDC 2255227 and 2255484 for the single-crystal structures of P$_4$@MFM-300(In) and(P$_8$)$_n$@MFM-300(In), respectively, and 2255485 for the structure of (P$_8$)$_n$@MFM-300(In) refined by the Rietveld Method. All data were available from the corresponding authors upon request. Source data are provided with this paper.

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

## Acknowledgements

We thank the EPSRC (EP/I011870, M.S.), the University of Manchester, the National Science Foundation of China, and Peking University for funding. This project has received funding from the European Research Council (ERC) under the European Union's Horizon 2020 research and innovation programme (grant agreement No 742401, NANOCHEM, M.S.). R.V.B. is grateful to colleagues at the Centre for Computational Materials Science and E-IMR Centre at the Institute for Materials Research, Tohoku University, for their support.

## Author contributions

S.A.S.: overall design of the project, synthesis and characterisation of materials, photocatalytic measurements, interpretation of spectroscopy data, preparation of the manuscript. R.V.B.: all quantum chemical calculations. I.J.V.-I., G.F.S.W., S.A.S.: single-crystal X-ray diffraction data collection and structure refinement, Y.C. and I.d.S.: collection and refinement of the powder diffraction data. X.C. and S.A.S.: electrochemical measurements of dielectric constants, experiments on electrochemical sensing of SO₂. S.A.S., J.M., L.S.N.: collection of Raman and luminescence spectra and determination of the luminescence lifetime. M.K.-J. and D.A.J.: gas adsorption measurements. C.B. and D.M.P.: collection of solid-state NMR spectra. V.P.F., S.Y., and M.S.: overall design, direction and supervision of the project, and preparation of the manuscript with all authors.

## Competing interests

The authors declare no competing interests.
