## [Transparent Peer Review file · Nature Communications]

Direct synthesis of a semiconductive double-helical phosphorus allotrope in a metal-organic framework

Corresponding Author: Professor Martin Schröder

Version 0:

Reviewer comments:

Reviewer #1

(Remarks to the Author)

The work described in the manuscript by Schroeder et al. is highly interesting and represents an important development in the field of P-based materials and MOF. Currently, there is also an increasing interest in the field of photocatalysis and this manuscript well contributes to this field, furnishing a new photocatalyst based on earth-abundant element as P, thus avoiding use of toxic metals or critical raw materials. Though the idea of absorption of P₄ in a MOF is not completely new, since there is in lit. the following work: Chem. Sci 2018, DOI: 10.1039/c8sc01439f, which I think should be cited in the introduction.

Regarding the X-ray characterization of the final product there are some aspects that are not clear to me: by the help of DFT that gives a model which is then used in the refinement of PXRD data, the authors conclude they have got a double helix structure inside the nanochannels of the MOF. But, as shown in Figure S6, the XRD pattern does not change going from the initial MOF to P₄@MOF and to P₈@MOF, what is the explanation of these data?

The methodology followed by the authors is sound, there is an extensive and in-depth characterization, though the paper lacks the XPS measurements to verify the oxidation state of P atoms inside the MOF. It would be interesting to compare a fresh sample of P_n@MFM-300(In) and a sample kept some time, for instance one month, in ambient conditions to verify the effective stability towards oxidation, which is Achille's hill for elemental P.

-In the description of the synthesis of P_n@MFM-300(In) there is no indication of the amount (mg or gr) of starting material MFM-300(In) and of the amount of the final isolated material. These data are useful for the reader to reproduce the experiment, thus they should be added. Moreover, the reader cannot estimate an approximate yield of the synthesis and there is no evaluation of the amount wt% of (P₈)_n encapsulated in the MOF. Please add the missing data. Additionally, it lacks the synthesis of the starting material MFM-300(In) or at least a citation of the literature where it is described. The authors say: "Activated guest-free MFM-300(In)" how do they carry out the activation of the starting material? Please add it.

-In the section Results and Discussion, 4th line it is written the MFM-300(In) powder is colourless, well I believe it is white, please correct it.

Reviewer #2

(Remarks to the Author)

The paper by Schöder and coworkers expose an interesting approach to control the growing of main-group clusters, employing "nano-reactors" or "nano-containers", favoring an unprecedented double-helical phosphorus array. The manuscript is well-written, and the novelty is clear. Thus, I suggest publication in the Nature Communication journal after considering the following small points:

#1

The authors may consider to get the solid-state ^{31}P -NMR patterns, and making a comparison to theoretical calculations of the free extended phosphorus-array, among other similar discrete species such as P14 and P16.

#2

Comments on the plausible dynamics of the P-skeleton are relevant.

#3

A tentative mechanism for the P4 polymerization can be of interest.

#4

The formation of multiple aromatic units within the extended P-array can be commented on the basis of the discussed work in literature, such as:

Dual Spherical Aromaticity in $[\text{M}(\mu_2\text{-P}_4)_2]^+$ ($\text{M}=\text{Cu}, \text{Ag}, \text{Au}$). Evaluation of Bonding Nature and Spherical Aromatic Character from Relativistic DFT Calculations. *Inorg. Chem. Comm.* 2024, 159, 111680. DOI: 10.1016/j.inoche.2023.111680

#5

Are the multiple P-P bonds equal, in terms of bond order, energy and nature?. Discuss inter, and intra-P8 units bonds.

#6

Discuss why the P4-structure is prone to thermal degradation, as seen from TGA measurements, in comparison to the P8-polymer.

#7

Discuss the tentative formation of other main-group chains within the MFM-300(In) MOF.

#8

How relevant is the In-metal center to the formation of the P8-polymeric structure? Recall to Hard-Soft Acid-Base (HSAB) theory.

#8

What about S8 monomeric species, which may lead to a similar situation? This can be a good source for upcoming work.

Reviewer #3

(Remarks to the Author)

The authors report an excellent piece of chemistry. They are capable to use a metal-organic framework as a chemical reactor to encapsulate white phosphorous within its channels and then to induce a conversion of P4 species into $(\text{P}_8)_n$. Remarkably, they are capable to follow the process with SCXRD. Remarkably, the resulting host-guest material, after the photo-polymerisation, exhibits remarkable semiconducting properties. Overall, the work is superb and, in my opinion, deserves publication in *Nat. Comm.*, as it is. I only recommend to introduce more citations related to the host-guest chemistry of MOFs (Fujita's work with crystalline sponge for instance or DOI: 10.1039/c8mh00302e for instance).

Reviewer #4

(Remarks to the Author)

The paper by Sapchenko, Yang, and Schröder presents significant findings on the synthesis, characterization, and potential applications of a new phosphorus allotrope stabilized within a metal-organic framework (MOF). The study is well-executed and provides valuable insights into the structure and properties of phosphorus allotropes, which could be of interest to researchers in materials science, chemistry, and nanotechnology. However, a few revisions are suggested to enhance the clarity, coherence, and overall quality of the manuscript.

1. I suggest rephrasing in the abstract as following: "This study provides the first crystallographic evidence of the structural building blocks of the red phosphorus allotrope, stabilized within the pores of MFM-300(In)."

2. The manuscript discusses the structural integrity and transformation of phosphorus within the MOF, but more detail on the SCXRD refinements corroborate the experimental findings would improve comprehension. Furthermore, additional characterization methods could be mentioned to further substantiate the findings. For example, if XPS, or other techniques were used to study the electronic environment or oxidation states of phosphorus, including this data would strengthen the results.

3. The results section could benefit from more explicit connections between the findings and their implications. For example, when discussing the Raman spectrum similarities between bulk red phosphorus and $(\text{P}_8)_n@ \text{MFM-300}(\text{In})$, it would be helpful to state why this finding is significant in confirming the presence of such helical chains in amorphous red phosphorus.

4. The section on the conversion of P4@MFM-300(In) to (P8)n@MFM-300(In) under irradiation could benefit from additional details. While it states the use of a "300 W Xe light source ($\lambda = 350\text{--}760\text{ nm}$)," specifying the intensity or flux of light and how this parameter was optimized for the polymerization process would add clarity. Were any controls or comparative experiments done with different wavelengths or intensities?

5. The discussion on the potential applications of the material as an air-stable semiconductor for gas detection and photocatalysis is interesting but somewhat limited. It would be beneficial to compare the performance of this new material with existing phosphorus-based semiconductors or other materials used in gas detection and photocatalysis. A brief comparison could help position this research within the broader context of material science applications.

6. There are a few typographical errors and awkward phrasings that should be corrected for clarity. For instance, "the all the P4 sites (I–III) give rise to a variety of possible positions of P4 within the channels of MFM-300(In)" should be rephrased to "all the P4 sites (I–III) result in a variety of possible positions of P4 within the channels of MFM-300(In)." Careful proofreading to catch such minor issues is recommended.

7. The conclusion briefly mentions the potential to extend this methodology to other porous materials. Consider elaborating on this point by providing specific examples of other MOFs or porous materials that could benefit from a similar approach, or discussing how different functional groups might influence phosphorus stabilization and reactivity.

8. While the manuscript frequently refers to the supplementary information (SI), ensure that all essential data and explanations required to understand the main text, if possible, are included within the paper itself. Important figures and tables that are integral to the discussion should not solely be relegated to the SI.

Version 1:

Reviewer comments:

Reviewer #1

(Remarks to the Author)

The revised manuscript is suitable for publication. All the Referee's requests have been carefully answered.

Reviewer #2

(Remarks to the Author)

The manuscript has been improved as suggested by the reviewer. Thus, I recommend publication in the current form.

Reviewer #4

(Remarks to the Author)

The authors have enhanced the paper by addressing additional requirements and providing further characterizations. I support its publication in its current form.

Reviewer #1

The work described in the manuscript by Schroeder et al. is highly interesting and represents an important development in the field of P-based materials and MOF. Currently, there is also an increasing interest in the field of photocatalysis and this manuscript well contributes to this field, furnishing a new photocatalyst based on earth-abundant element as P, thus avoiding use of toxic metals or critical raw materials. Though the idea of absorption of P₄ in a MOF is not completely new, since there is in lit. the following work: Chem. Sci 2018, DOI: 10.1039/c8sc01439f, which I think should be cited in the introduction.

We thank to the reviewer for their comments. We have added the reference to the paper as Ref 21

Regarding the X-ray characterization of the final product there are some aspects that are not clear to me: by the help of DFT that gives a model which is then used in the refinement of PXRD data, the authors conclude they have got a double helix structure inside the nanochannels of the MOF. But, as shown in Figure S6, the XRD pattern does not change going from the initial MOF to P₄@MOF and to P₈@MOF, what is the explanation of these data?

The PXRD patterns in Figure S6 were recorded using conventional low-resolution technique. The most intense and characteristic peaks visible at low angle are for the MOF framework. This is why their position and intensity remains almost unchanged upon inclusion and polymerisation of P₄. But if we compare the groups of peaks at $2\theta = 27-33^\circ$ and $35-40^\circ$ in PXRD plots of P₄@MFM-300(In) and (P₈)_n@MFM-300(In) with bare MFM-300(In), there are changes in the intensity and position of peaks, reflecting the presence of guest molecules. This is demonstrated by Rietveld refinement of the high-resolution diffractogram (Fig. S7), high resolution being required to resolve such structural details.

The methodology followed by the authors is sound, there is an extensive and in-depth characterization, though the paper lacks the XPS measurements to verify the oxidation state of P atoms inside the MOF. It would be interesting to compare a fresh sample of P_n@MFM-300(In) and a sample kept some time, for instance one month, in ambient conditions to verify the effective stability towards oxidation, which is Achille's hill for elemental P.

We are thankful for this suggestion. Although XPS measurements could be useful, Raman spectroscopy is very efficient and effective for the characterisation of various forms of P in these materials (<https://doi.org/10.1070/RC1961v030n07ABEH002989>), and in our case has been used to verify the stability of the material towards oxidation. P–O vibrations are usually observed within the range of $850-1000\text{ cm}^{-1}$ (<https://doi.org/10.1016/j.saa.2013.01.008>, DOI:10.1180/0026461026660077, DOI: 10.1002/jrs.1253). Indeed, the spectrum of fresh P_n@MFM-300(In) shows a very weak peak at 924 cm^{-1} which is not due to framework vibrations, and indicates the presence of P–O bonds. The Raman spectrum of a 3 month-old sample is almost identical to that of the original fresh phase, and the signal at 924 cm^{-1} remains weak, suggesting that no oxidation takes place over this timescale. Given the consistently low intensity of the peak at 924 cm^{-1} we argue that only some surface oxidation takes place, and that the bulk material is stable. We have added some discussion to the manuscript on the stability at the end of the 1st paragraph of the Spectroscopy section. The relevant spectra have been added to Section 7 of ESI (Fig. S14b).

In the description of the synthesis of P_n@MFM-300(In) there is no indication of the amount (mg or gr) of starting material MFM-300(In) and of the amount of the final isolated material. These data are useful for the reader to reproduce the experiment, thus they should be added.

Moreover, the reader cannot estimate an approximate yield of the synthesis and there is no evaluation of the amount wt% of (P₈)_n encapsulated in the MOF. Please add the missing data.

We have added the amounts of starting material and reaction yield to the methods section. The concentration of phosphorus in the polymerised sample is now given in the main text in the section Conversion of P₄@MFM-300(In) to (P₈)_n@MFM-300(In) (paragraph 1, line 5).

Additionally, it lacks the synthesis of the starting material MFM-300(In) or at least a citation of the literature where it is described. The authors say: "Activated guest-free MFM-300(In)" how do they carry out the activation of the starting material? Please add it.

We apologise for the confusion. The literature reference to the synthesis of MFM-300(In) was provided in the Supporting Information (ref. 1). Following the above advice we have added the description of our synthesis of MFM-300(In) to the Methods section and have quoted the original reference as ref 22.

In the section Results and Discussion, 4th line it is written the MFM-300(In) powder is colourless, well I believe it is white, please correct it.

We have rephrased.

Reviewer #2

The paper by Schöder and coworkers expose an interesting approach to control the growing of main-group clusters, employing "nano-reactors" or "nano-containers", favoring an unprecedented double-helical phosphorus array. The manuscript is well-written, and the novelty is clear. Thus, I suggest publication in the Nature Communication journal after considering the following small points:

#1

The authors may consider to get the solid-state ^{31}P -NMR patterns, and making a comparison to theoretical calculations of the free extended phosphorus-array, among other similar discrete species such as P_{14} and P_{16} .

We are thankful to the reviewer for this suggestion. We have now collected solid-state ^{31}P -NMR spectra of bulk red phosphorus and of $\text{P}_n\text{@MFM-300(In)}$ materials and added these data to a new Section 8 in ESI. We also modified the Spectroscopy part in the main text of the article. However, the detailed interpretation of these NMR spectra remains challenging. This remains the focus of future work in this area.

#2

Comments on the plausible dynamics of the P-skeleton are relevant.

Although indeed intriguing, the dynamics of the phosphorus double chain have not been investigated. Due to the inherent complexity of the system, the study on the dynamic behaviour of the encapsulated polymer will require extensive use of molecular dynamics simulations with multiple experimental methods such as variable temperature NMR spectroscopy. This will be the direction of future work.

#3

A tentative mechanism for the P_4 polymerization can be of interest.

Establishing the mechanism of such a complex process will require a series of additional experiments and calculations which are well beyond the scope of this paper. Speculations on plausible mechanisms can be made, but these would be based on structural data only and therefore would be overly speculative for inclusion into the manuscript. We have though added additional text into paragraph 1 in the section on Conversion of $\text{P}_4\text{@MFM-300(In)}$ to $(\text{P}_8)_n\text{@MFM-300(In)}$ relating to structural details. We are really anxious not to speculate too much on mechanisms when we do not have full data or evidence to support any proposed mechanism.

#4

The formation of multiple aromatic units within the extended P-array can be commented on the basis of the discussed work in literature, such as: Dual Spherical Aromaticity in $[\text{M}(\mu_2\text{-P}_4)_2]^+$ ($\text{M}=\text{Cu}, \text{Ag}, \text{Au}$). Evaluation of Bonding Nature and Spherical Aromatic Character from Relativistic DFT Calculations. Inorg. Chem. Comm. 2024, 159, 111680. DOI: 10.1016/j.inoche.2023.111680

We are thankful to the reviewer for highlighting this work. Yes, the aromatisation of P_4 fragment is possible upon coordination, as we mention in the section on Conversion of $\text{P}_4\text{@MFM-300(In)}$ to $(\text{P}_8)_n\text{@MFM-300(In)}$ part (ref. 31-34). We have added the above paper mentioned by the reviewer as an additional example of spherical aromaticity (ref. 34). Regarding the P_8 chains, despite some flattening of the P_4 butterfly units within P_8 cluster and relatively short interatomic distances, the fragment remains bent and the double helix is twisted. This confirms the absence of classical

aromaticity in the Hückel sense. The paper by Ulloa et al. gives some insight on this question: they demonstrate that contrary to $[M(\eta^2-P_4)_2]^+$ complexes, a hypothetical phosphorus cage $[P(P_4)_2]^+$ should lose spherical aromaticity and would result in significant shifts of the phosphorus signal in ^{31}P NMR spectra: from -522 ppm for the case of aromatic P_4 cluster to -247.5 ppm. The observed peak in the solid state ^{31}P NMR spectrum of $(P_8)_n@MFM-300(In)$ (-120 ppm) is even more shifted, which is consistent with the absence of spherical aromaticity.

#5

Are the multiple P-P bonds equal, in terms of bond order, energy and nature?. Discuss inter, and intra-P8 units bonds.

The bond distances demonstrate some variation as shown on Fig S5. The P–P bond lengths within comparable with P–P bond length reported for sterically-hindered diphosphanes [2.291(4)–2.357(2) Å]

s

]

,

We have added this text to paragraph 1 in the section Conversion of $P_4@MFM-300(In)$ to $(P_8)_n@MFM-300(In)$ to highlight this question.

Discuss why the P4-structure is prone to thermal degradation, as seen from TGA measurements, in comparison to the P8-polymer.

P_4 is volatile and evaporates from the sample of $P_4@MFM-300(In)$ upon heating. The first weight loss step as indicated in the Methods section exactly matches with loss of all guest P_4 molecules. $(P_8)_n$ in $(P_8)_n@MFM-300(In)$ is much less volatile and therefore weight loss for $P_4@MFM-300(In)$ occurs at lower temperatures due to desorption of water and decomposition profile looks different to $P_4@MFM-300(In)$. We highlight these differences and have added text to the caption of the TGA plots (ESI, Fig. S2).

#7

Discuss the tentative formation of other main-group chains within the MFM-300(In) MOF.

S, Se, As and Sb seem to be the most prominent candidates as their molecular forms can be evaporated at reasonably low temperatures. We have added some hints on their use in the end of Summary section but this topic is beyond the scope of the present work.

#8

How relevant is the In-metal center to the formation of the P8-polymeric structure? Recall to Hard-Soft Acid-Base (HSAB) theory.

The most populated and energetically favourable site for phosphorus adsorption in $P_4@MFM-300(In)$ is the OH-group coordinated to In. The metal polarises the OH group, which should strengthen electrostatic and hydrogen bonding interactions between OH-groups and guest phosphorus molecules. However, upon the formation of the polymer, the chains drift away from OH-groups. The phosphorus atoms still interact with oxygens from carboxylate groups, which are also coordinated to Indium. In this sense, the metal indeed plays a key role in the binding of phosphorus atoms by the framework. However, PDOS calculations demonstrate minimal if any contribution of indium orbitals to the band gap region (Fig. S12) which means the metal orbitals do not play a significant role in the emergence of semiconductive properties of the material.

#9

What about S8 monomeric species, which may lead to a similar situation? This can be a good source for upcoming work.

We are thankful for this suggestion. We have interests in encapsulation of sulphur allotropes in the context of materials for batteries, but this is beyond the scope of the present paper.

Reviewer #3 (Remarks to the Author):

The authors report an excellent piece of chemistry. They are capable to use a metal-organic framework as a chemical reactor to encapsulate white phosphorous within its channels and then to induce a conversion of P₄ species into (P₈)_n. Remarkably, they are capable to follow the process with SCXRD. Remarkably, the resulting host-guest material, after the photo-polymerisation, exhibits remarkable semiconducting properties.

Overall, the work is superb and, in my opinion, deserves publication in Nat. Comm., as it is. I only recommend to introduce more citations related to the host-guest chemistry of MOFs (Fujita's work with crystalline sponge for instance or DOI: 10.1039/c8mh00302e for instance).

We thank the reviewer for their generous comments. We have added the above reference and a more recent one (refs. 18, 19).

Reviewer #4 (Remarks to the Author):

The paper by Sapchenko, Yang, and Schröder presents significant findings on the synthesis, characterization, and potential applications of a new phosphorus allotrope stabilized within a metal-organic framework (MOF). The study is well-executed and provides valuable insights into the structure and properties of phosphorus allotropes, which could be of interest to researchers in materials science, chemistry, and nanotechnology. However, a few revisions are suggested to enhance the clarity, coherence, and overall quality of the manuscript.

1. I suggest rephrasing in the abstract as following: "This study provides the first crystallographic evidence of the structural building blocks of the red phosphorus allotrope, stabilized within the pores of MFM-300(In)."

The abstract has been amended accordingly.

2. The manuscript discusses the structural integrity and transformation of phosphorus within the MOF, but more detail on the SCXRD refinements corroborate the experimental findings would improve comprehension. Furthermore, additional characterization methods could be mentioned to further substantiate the findings. For example, if XPS, or other techniques were used to study the electronic environment or oxidation states of phosphorus, including this data would strengthen the results.

Although XPS measurements could be useful, Raman spectroscopy is very efficient and effective for the characterisation of various forms of P (<https://doi.org/10.1070/RC1961v030n07ABEH002989>), and in our case has been used to verify the stability of the material towards oxidation. P–O vibrations are usually observed within the range of 850-1000 cm⁻¹ (<https://doi.org/10.1016/j.saa.2013.01.008>, DOI:10.1180/0026461026660077, DOI: 10.1002/jrs.1253). Indeed, the spectrum of fresh P_n@MFM-300(In) shows a very weak peak at 924 cm⁻¹ which is not due to framework vibrations, and therefore indicates the presence of P–O bonds. The Raman spectrum of a 3 month-old sample is identical to the original fresh phase, and the signal at 924 cm⁻¹ remains weak, which suggests that no oxidation takes place over this timescale. Given the consistently low intensity of the peak at 924 cm⁻¹ we argue that only some surface oxidation takes place, but this does not lead to a full decomposition of the material. We have added some discussion to the manuscript on the stability at the end of the 1st paragraph of the Spectroscopy section. The relevant spectra have been added to Section 7 of ESI (Fig. S14b).

3. The results section could benefit from more explicit connections between the findings and their implications. For example, when discussing the Raman spectrum similarities between bulk red phosphorus and (P₈)_n@MFM-300(In), it would be helpful to state why this finding is significant in confirming the presence of such helical chains in amorphous red phosphorus.

This has been added.

4. The section on the conversion of P₄@MFM-300(In) to (P₈)_n@MFM-300(In) under irradiation could benefit from additional details. While it states the use of a "300 W Xe light source (λ = 350–760 nm),"

specifying the intensity or flux of light and how this parameter was optimized for the polymerization process would add clarity. Were any controls or comparative experiments done with different wavelengths or intensities?

We have added the missing information to Section 1 of ESI (paragraph 1, line 10). Since the yield of the reaction was close to quantitative, we did not seek to study the effect of changes to the parameters of irradiation.

5. The discussion on the potential applications of the material as an air-stable semiconductor for gas detection and photocatalysis is interesting but somewhat limited. It would be beneficial to compare the performance of this new material with existing phosphorus-based semiconductors or other materials used in gas detection and photocatalysis. A brief comparison could help position this research within the broader context of material science applications.

To make the comparison easier we have gathered information on the performance of phosphorus-based materials and MOFs for photodegradation of dyes. This is a new Table 1 in the main manuscript.

6. There are a few typographical errors and awkward phrasings that should be corrected for clarity. For instance, “the all the P4 sites (I–III) give rise to a variety of possible positions of P4 within the channels of MFM-300(In)”* should be rephrased to “all the P4 sites (I–III) result in a variety of possible positions of P4 within the channels of MFM-300(In).” Careful proofreading to catch such minor issues is recommended.

We are thankful for the careful reading of our manuscript. We have proofread the manuscript and made changes as appropriate.

7. The conclusion briefly mentions the potential to extend this methodology to other porous materials. Consider elaborating on this point by providing specific examples of other MOFs or porous materials that could benefit from a similar approach, or discussing how different functional groups might influence phosphorus stabilization and reactivity.

The most probable hosts for polymerisation reactions are other MOFs featuring 1D channels. Since there is a strong interaction of guest phosphorus molecules with OH-groups coordinated to metal, it would be interesting to investigate their influence on the phosphorus polymerisation in other OH-containing MOFs,. The summary has been suitably extended.

8. While the manuscript frequently refers to the supplementary information (SI), ensure that all essential data and explanations required to understand the main text, if possible, are included within the paper itself. Important figures and tables that are integral to the discussion should not solely be relegated to the SI.

We have added more discussion on the topics highlighted by the reviewers and moved some material from the electronic supporting information into the methods and discussion sections of the manuscript, including a new Table 1. We have also added a reference to luminescence properties of the material in the main text (Spectroscopy section, paragraph 3, line 3). We are though sensitive to the word limits set by the journal.